# Preserving Esthetics: Interventional Radiotherapy (Brachytherapy) as a Potential Alternative to Surgery for Basal Cell Carcinoma of the Midface

**DOI:** 10.3390/jcm14103305

**Published:** 2025-05-09

**Authors:** Alessia Re, Sebastiana Lai, Glenda Mantione, Andrea D’Aviero, Fabrizio Sanna, Elisa Pilloni, Sebastiano Menna, Danila Piccari, Althea Boschetti, Bruno Fionda, Davide Porru, Pierangela Tramaloni, Roberto Gallus, Luca Tagliaferri, Maria Antonietta Montesu, Corrado Rubino, Francesco Bussu, Gian Carlo Mattiucci

**Affiliations:** 1Radiation Oncology, Mater Olbia Hospital, 07100 SassariOlbia, Italy; alessia.re@materolbia.com (A.R.); elisa.pilloni@materolbia.com (E.P.); sebastiano.menna@materolbia.com (S.M.); danila.piccari@materolbia.com (D.P.); althea.boschetti@medipass.it (A.B.); giancarlo.mattiucci@unicatt.it (G.C.M.); 2Division of Otolaryngology, Azienda Ospedaliera Universitaria, 07100 Sassari, Italy; sebastiana.lai@aouss.it (S.L.); glendamantione1@gmail.com (G.M.); pierangela.tramaloni@aouss.it (P.T.); 3Department of Medicine, Surgery and Pharmacy, University of Sassari, 07100 Sassari, Italy; mmontesu@uniss.it (M.A.M.); corubino@uniss.it (C.R.); 4Department of Medical, Oral and Biotechnological Sciences, “G. D′Annunzio” University of Chieti, 66100 Chieti, Italy; andreadav89@gmail.com; 5Department of Radiation Oncology, “S.S. Annunziata” Chieti Hospital, 66100 Chieti, Italy; 6Division of Radiotherapy, Azienda Ospedaliera Universitaria, 07100 Sassari, Italy; fabriziosannasassari@yahoo.it; 7Dipartimento di Diagnostica per Immagini, Radioterapia Oncologica ed Ematologia, Policlinico Universitario “A. Gemelli” IRCCS, 00100 Rome, Italy; bruno.fionda@policlinicogemelli.it (B.F.); luca.tagliaferri@policlinicogemelli.it (L.T.); 8Otolaryngology, Mater Olbia Hospital, 07026 Olbia, Italy; davide.porru@materolbia.com (D.P.); roberto.gallus@materolbia.com (R.G.); 9Radiotherapy Department, Università Cattolica del Sacro Cuore, 00100 Roma, Italy; 10Dermatology Division, Azienda Ospedaliera Universitaria, 07100 Sassari, Italy; 11Division of Plastic Surgery, Azienda Ospedaliera Universitaria, 07100 Sassari, Italy

**Keywords:** skin cancer, basocellular carcinoma, interventional radiotherapy, brachytherapy, quality of life, brachytherapy implants, midface tumors

## Abstract

**Background/Objectives**: High-dose-rate (HDR) interventional radiotherapy (IRT) represents a valuable treatment alternative for basal cell carcinoma (BCC) of the midface, especially in the presence of esthetic or functional issues related to surgery. This retrospective study examines the clinical outcomes of patients who underwent HDR-IRT at our institution. **Methods**: Patients (pts) with histologically confirmed midface BCC were treated with HDR-IRT using superficial, interstitial, or endocavitary implants depending on tumor characteristics. Fractionation regimens were tailored to tumor location: 45 Gy in 9 fractions twice a day (the cheeks/lips) and 44 Gy in 14 fractions (the nose, with the first and last fractions delivering 4 Gy each, administered once daily, and the remaining 12 fractions delivering 3 Gy each, administered twice daily (BID) with an inter-fraction interval of at least 6 h). Treatment efficacy was assessed based on local control rates, toxicity (CTCAE criteria), and cosmetic outcomes. **Results**: Eight patients were considered. The most common tumor site was the nose (seven patients, 87.5%), followed by the upper lip. Tumors were either primary (three patients, 37.5%) or residual/recurrent after previous surgery with involved margins (five patients, 62.5%). The median follow-up was 6 months (range: 1–19 months). Clinical local control was achieved in all cases, with persistent alteration at dermoscopy in one patient six months after the completion of treatment. Acute toxicities were minimal, with two patients developing grade 3 skin toxicity, which resolved within 1–3 months with topical management. Cosmetic outcomes were favorable across all patients. **Conclusions**: HDR-IRT is a well-tolerated, effective, and cosmetically favorable treatment for midface BCC. Our institutional experience supports its use as an alternative to surgery in cosmetically sensitive areas and in selected patients/lesions unfit for surgery. Endocavitary implants appear to be an effective option to cover the full thickness nasal wall without resorting to interstitial implants.

## 1. Introduction

Basal cell carcinoma (BCC) is the most common malignant skin tumor, characterized by locally invasive growth and an extremely low risk of metastasis (0.5–0.1% of cases). It primarily affects individuals over the age of 40, with a predilection for chronically sun-exposed areas of the body, particularly the face. Accounting for approximately 15% of all malignancies, BCC represents a significant public health concern [1,2].

Surgical excision, both “conventional” and with Peripheral Deep En Face Margin Assessment (PDEMA), represents the first-choice treatment, offering high cure rates and long-term disease control. However, surgery may not always be feasible, particularly in elderly patients (pts) or those with significant comorbidities, where the risks associated with anesthesia and postoperative complications must be carefully considered. Moreover, when BCC arises in cosmetically sensitive areas, especially on the face, surgical intervention can lead to significant esthetic sequelae, potentially causing substantial disfigurement and impacting patients’ quality of life, and it may require complex reconstructive procedures.

These concerns are particularly evident in the area which includes the nose, superior lip, and labial rim, which has been previously defined as the midface [3]. In the nose, for example, the cartilage framework creates subtle, three-dimensional contours that are covered by very thin skin. This delicate structure means that even minimal irregularities in the reconstructed relief or hollows can be easily detected by natural lighting and shadows, while also tending to be immediately noticed by observers. In addition, the superior lip and labial rim are not just cosmetic landmarks but also critical to oral function with different yet equally complex reconstructive issues. The psychological impact of disfigurement in the midface area can be profound because any imperfection in these highly visible areas can affect a patient’s self-image and quality of life.

Given that BCC is a malignancy with an excellent prognosis in the vast majority of cases, minimizing treatment-related morbidity while maintaining efficacy is a crucial concern. Therefore, the search for effective, organ-preserving alternatives to surgery is of paramount importance. In this context, high-dose-rate interventional radiotherapy (HDR-IRT) emerges as a promising option, providing high local control rates while offering the advantage of superior cosmetic outcomes compared to surgical excision [4].

HDR-IRT for non-melanoma skin cancers (NMSCs), including BCC, has gained renewed interest due to its ability to deliver highly conformal radiation while sparing adjacent healthy tissues. The American Brachytherapy Society and the GEC-ESTRO group, based upon current evidence, have provided guidelines supporting the use of HDR brachytherapy for skin cancers, emphasizing its advantages in terms of local control, toxicity, and cosmetic outcomes [5,6].

Despite its benefits, HDR-IRT for BCC remains heterogeneous in terms of dose prescription, fractionation, and treatment techniques. Variability in clinical practice is evident in the published literature, with fractionation regimens ranging from 6 to more than 30 fractions and total doses varying between 30 and 74 Gy depending mainly on tumor characteristics, location, and types of implants [5].

This study presents our institutional experience with HDR-IRT for midface BCC, analyzing treatment outcomes, recurrence rates, toxicity profiles, and cosmetic results, and we compare our findings with the latest evidence in the literature.

This work describes our experience, discussing it under the light of analogous series in the literature, aiming to highlight the efficacy, safety, and future directions of HDR-IRT in managing midface BCC.

## 2. Materials and Methods

This retrospective study analyzes the clinical outcomes of patients treated with HDR-IRT for BCC of the midface at our institution.

All cases were evaluated by a dermatologist through dermoscopy, and they were reviewed and discussed by the Multidisciplinary Tumor Board of Northern Sardinia of Head and Neck cancers before treatment initiation to confirm diagnosis, assess tumor characteristics, and determine eligibility for IRT.

The type of implant was determined based on the characteristics of the lesion. For superficial lesions (<5 mm depth), a superficial implant for contact brachytherapy was considered sufficient. In cases of larger tumors, aggressive histology, or targets located on curved surfaces or in proximity to organs at risk, where deeper dose delivery was required, interstitial or endocavitary implants (using intranasal packing for dose application) were considered either in addition to or as an alternative to superficial catheters.

Patients included in the study had histologically confirmed midface BCC, either as a primary lesion or as a recurrence/residual after previous treatments such as surgery or other therapies (e.g., cryotherapy), where surgery was associated with concerns related to cosmetic or functional considerations. Clinical history, pathology, and patient demographics were collected.

All patients were treated at our Radiation Oncology Department, Mater ART (Advanced Radiation Therapy), located in Olbia, Sardinia. In cases requiring the placement of interstitial or endocavitary catheters, the procedure was performed at the Otolaryngology Division in Sassari.

HDR-IRT was delivered using the Bravos^®^ (Varian) system, utilizing an Iridium-192 radioactive source.

For superficial implants, treatment was performed using plastic tubes positioned over a bolus, which was placed over a thermoplastic mask to ensure reproducibility and dose conformity. In this case, the catheters are placed in parallel along the area to be irradiated.

When deeper dose delivery is required, endocavitary implants (using an intranasal packing for dose application) and interstitial implants (with the final geometry of the implants being primarily dictated by the anatomic subchondral planes) [7,8] are considered. In every case, the exact configuration and number of catheters are individually tailored to the extent, depth, and shape of the target lesions. In Figure 1, all three types of implants are shown.

The target region was delineated during the simulation phase using radiopaque markers based on the physical examination, surgical scars, and photographs provided by the patients showing the lesion prior to surgery.

The delineation of the Clinical Target Volume (CTV) was performed jointly by a radiation oncologist, a dermatologist, and an otorhinolaryngologist, ensuring precise coverage of the tumor while minimizing radiation exposure to surrounding healthy tissues.

For all types of implants, three-dimensional implant reconstructions were required. Manual dwell time adjustments were performed to personalize dose homogeneity and to avoid skin hotspots.

Two distinct fractionation regimens were employed based on tumor location:For tumors located on the cheeks and lips, patients received 45 Gy in 9 fractions, delivered twice in a day (bis in die, BID), with at least 6 h between doses.For tumors on the nose surface, a fractionation scheme of 44 Gy in 14 fractions was used. This regimen consisted of the following:
The first and last fractions delivering 4 Gy each, administered once daily;The remaining 12 fractions delivering 3 Gy each, administered BID with an inter-fraction interval of at least 6 h.

Patients were followed-up with at regular intervals to assess local control, toxicity, and cosmetic outcomes. Cosmetic outcomes were assessed during routine clinical follow-up visits by either the otolaryngologist or the radiation oncologist. Evaluation was based on direct visual inspection and palpation of the treated area, taking into account skin texture, pigmentation, telangiectasia, and contour irregularities. Local control was defined as the clinical absence of recurrence at the treatment site, while toxicity was evaluated using Common Terminology Criteria for Adverse Events (CTCAEs) v5.0 [9]. Further evaluation was performed by the dermatologist through dermoscopy. Cosmetic results were assessed based on physician and patient evaluations.

## 3. Results

A total of eight patients were included in the study, with a median age of 71 years (range: 57–90 years). The most common tumor site was the nose (seven patients, 87.5%), followed by the upper lip. Tumors were either primary (three patients, 37.5%) or recurrent/residual (five patients, 62.5%). Most recurrent/residual cases had previously undergone several surgical excisions, with some patients receiving additional treatments such as cryotherapy (Table 1).

The mean CTV was 10.8 cc (range: 1.08–23.28 cc). The majority of patients were treated with 14 fractions (87.5%), while one patient received 9 fractions. The number of plastic tubes used per patient ranged from 4 to 11. Endocavitary tubes were used in five cases (62.5%). Interstitial tubes were required in two (25%) patients, namely the one with the upper lip disease and another patient with a nose recurrence arising at the level of surgical reconstruction (Table 2).

The median follow-up duration was 6 months (range: 1–19 months). At the last follow-up, the local control rate was 100%. One patient (12.5%) presented persistent alterations at dermoscopy 6 months after treatment without clinical evidence of disease.

Toxicities during treatment were limited, with no severe adverse effects reported. At least two patients developed acute grade 3 skin toxicity, which appeared approximately one week after treatment completion and was characterized by moist desquamation and significant erythema. Management included topical dressings with hyaluronic acid (sodium hyaluronate), antiseptic agents, and antibiotic creams, leading to complete resolution within 1 to 3 months.

The cosmetic outcomes were favorable in all patients. At the last follow-up, three out of eight patients presented with mild nasal retraction, while one patient exhibited limited telangiectatic changes within the irradiated skin field. These alterations appeared gradually over time, becoming clinically appreciable approximately 6–12 months post-treatment, and they remained stable thereafter. All cosmetic evaluations were performed during routine clinical follow-up by either the otolaryngologist or radiation oncologist based on direct visual inspection and palpation of the treated area. No formal patient-reported outcome measures were employed. Despite this limitation, the clinical assessment suggests that these late effects were subtle and did not result in significant esthetic or functional impairments (Figure 2).

## 4. Discussion

Our findings confirm that HDR-IRT is an effective and well-tolerated treatment for basal cell carcinoma (BCC) of the face, in line with previous studies reporting local control rates above 90% [10]. This aligns with results from the large international meta-analysis conducted by Lee et al., which reported that brachytherapy achieved a 1-year local recurrence rate of 0% (95% CI, 0–0.5%) [11].

In our cohort, local control was achieved in 87.5% of cases, which is consistent with the existing literature. Laliscia et al. reported a 2-year local control rate of 90% in patients with head and neck non-melanoma skin cancers—primarily BCC—treated with 192Ir-based HDR brachytherapy using customized surface molds [12]. Tagliaferri et al. reported a 94% local control rate in facial lesions treated with contact HDR-IRT, along with highly favorable cosmetic outcomes [13]. Similarly, Chicheł et al. observed a complete response in all elderly patients with facial BCC treated with personalized HDR brachytherapy, further supporting the efficacy of this approach [14].

Further supporting this evidence, Krzysztofiak et al. reported a 100% clinical response in a cohort of 39 patients with high-risk BCCs of the midface region treated with 45 Gy in nine fractions of HDR contact brachytherapy using custom-made molds. All patients achieved complete remission, and cosmetic outcomes were rated as good or excellent in every case, further validating the high efficacy and esthetic benefit of HDR-IRT in delicate anatomical subunits [15].

Our results can also be compared with those of Yadavalli et al., who retrospectively evaluated 15 patients with non-melanoma skin cancer—including nine cases of basal cell carcinoma—treated with 3D-based HDR surface mold brachytherapy (SMBT). In their series, lesions were predominantly located on the face and head, and patients received a median total dose of 40 Gy in 10 fractions, administered using Iridium-192 and individualized CT-based planning. A complete response rate of 90% was reported among definitive cases, with no grade 3–5 toxicities in the majority of patients. In contrast, our cohort included only midface basal cell carcinomas, most of which were located on the nose, and they were treated using a broader range of implant techniques—including superficial, interstitial, and endocavitary approaches—according to tumor location and depth. Additionally, we adopted hypofractionated regimens, tailored to the treated subsite, with excellent local control and toxicity profiles [16].

The SCRiBE meta-analysis conducted by Zaorsky et al. revealed that HDR brachytherapy had lower recurrence rates and better cosmetic outcomes compared to external beam radiotherapy (EBRT), supporting its preferential use for small BCC lesions [17]. Skin brachytherapy is an excellent option to preserve function and cosmesis, especially in skin cancers located in the head and neck region [18].

Cosmesis is a critical consideration in the treatment of facial basal cell carcinoma (BCC), and our results indicate that HDR-IRT provides excellent esthetic outcomes with minimal late toxicities. These results are reinforced by the findings obtained by Lee et al., who, in their meta-analysis, observed that brachytherapy provided the highest rate of “good” cosmesis among all modalities—97.6%—significantly better than EBRT (74.6%) [11].

Although no formal patient-reported outcome measures were employed, esthetic outcomes were consistently evaluated by experienced clinicians through direct visual and tactile examination during follow-up visits, providing a reliable estimation of cosmetic results. In our cohort, the majority of patients achieved good or excellent cosmetic results, consistent with previous reports. Shah et al. found that 95% of patients experienced favorable esthetic outcomes following treatment [5], while Tagliaferri et al. reported poor cosmetic results in only 7% of treated lesions [13]. Similarly, Chicheł et al. reported that even in frail patients with advanced facial BCCs, individualized HDR brachytherapy allowed for effective tumor control while preserving both function and acceptable cosmetic outcomes [14]. These findings are consistent with those of Laliscia et al., who reported excellent cosmetic outcomes in 65% of lesions and no grade ≥ 3 acute or late toxicities in their cohort of patients treated with individualized HDR brachytherapy for head and neck skin cancers [12], further reinforcing the role of HDR-IRT as a cosmetically favorable and safe therapeutic option.

In our cohort, at least two patients developed acute grade 3 skin toxicity, which manifested approximately one week after the completion of treatment. These patients presented with moist desquamation and significant erythema, requiring intensive local wound care. Consistently, Laliscia et al. reported only grade 1–2 acute toxicities—most commonly mild dermatitis (52%), pain (25%), and ulceration (22%)—with no instances of grade ≥3 events. Late toxicities in their series were also mild, limited to grade 1 fibrosis (17%), atrophy (15%), and hypopigmentation (12%) [12]. Similarly, Krzysztofiak et al. described frequent low-grade dermatitis and a single case of grade 4 ulceration that completely resolved within six weeks [15]. These findings, including ours, reinforce the favorable tolerability profile of HDR-IRT, where acute toxicities, though not uncommon, are typically self-limiting and manageable with appropriate supportive care.

Treatment consisted of topical dressings with hyaluronic acid, antiseptic agents, and antibiotic cream to prevent secondary infections. The skin reactions gradually resolved over a period ranging from 1 to 3 months, with no evidence of long-term fibrosis or persistent ulceration. This is consistent with the low rates of late toxicities reported in the literature. Krzysztofiak et al. noted that the most common acute adverse effect was radiodermatitis, but late toxicities were rare and typically limited to hypopigmentation and mild atrophy [19].

The occurrence of grade 3 acute toxicity can be associated with the hypofractionated nature of our treatment regimen. Hypofractionation, characterized by higher doses per fraction, is known to increase the risk of acute skin reactions due to the reduced capacity for sublethal damage repair between fractions. According to the linear-quadratic model, higher doses per fraction result in an increased equivalent dose in 2 Gy fractions (EQD2), which correlates with a greater biological effect on normal tissues. In particular, fractionation schedules exceeding 3 Gy per session have been linked to a higher incidence of moist desquamation regimens [20].

However, the complete resolution of these adverse effects within a maximum of 3 months supports the idea that acute toxicity in HDR-IRT is often self-limiting and manageable with appropriate supportive care. This choice is consistent with the strategy proposed by Tagliaferri et al., who adopted accelerated hypofractionated schedules (e.g., 40–54 Gy in 8–18 fractions) for facial BCCs located in high-risk esthetic areas [13], and by Chicheł et al., who demonstrated the feasibility and excellent tolerability of personalized HDR brachytherapy in frail elderly patients with locally advanced BCCs [14]. Notably, the complete resolution of these adverse effects within a maximum of 3 months supports the notion that acute toxicity in HDR-IRT is generally self-limiting and manageable with appropriate supportive care, as also reported by Guinot et al. [21] and Rodríguez et al. [22], who observed that acute reactions following hypofractionated HDR-IRT are typically transient and do not result in long-term functional impairment.

Along the same line, Tang et al. reported that even in a cohort treated with personalized HDR electronic brachytherapy—including some cases receiving total doses of up to 80 Gy—acute toxicities mostly remained at grade 1, with no grade 4 events and excellent cosmetic outcomes throughout. Their findings underscore both the safety of hypofractionation and the value of tailored dose escalation for specific subtypes, such as pigmented BCCs [23].

These findings suggest that while hypofractionation may increase the incidence of acute grade 3 toxicity, it remains an effective strategy for optimizing treatment duration and patient compliance, particularly in elderly individuals or those with comorbidities. Future studies could explore dose modulation strategies or skin-protective interventions, such as advanced hydrocolloid dressings or prophylactic barrier creams, to further minimize the risk of acute adverse effects while maintaining high local control rates.

The fractionation regimens used in HDR-IRT vary significantly, as highlighted by the systematic review by Krzysztofiak et al. [19], which showed that local control rates ranged from 71% to 99% across different fractionation schedules [19]. The recommended biologically effective dose (EQD2-10) for BCC is 56 Gy or higher [5], with more protracted fractionation schedules being preferred for larger lesions or those in cosmetically sensitive areas.

Comparing our most used dose regimen (44 in 14 fractions) to the findings in the literature, we observe that it is slightly lower than the range of regimens reported in major studies [5,6].

This regimen represents the most frequently used fractionation scheme at our institution, particularly for lesions located in the vestibular mucosa. The rationale behind this fractionation schedule is based on tumor control optimization and toxicity reduction, balancing the benefits of hypofractionation while avoiding excessive late effects.

Our regimen consists of an initial and final fraction of 4 Gy and twelve fractions of 3 Gy BID, which yields an EQD2 of 48.18 Gy.

Tagliaferri et al. used 40 Gy in 8 fractions, with the BID scheme, resulting in an EQD2 of 50 Gy; in other protocols, 54 Gy was used, with 3 Gy/fraction with a regimen of 3 days/week (EQD2 58.5 Gy) [13], which is higher than our regimen, but it has a longer Overall Treatment Time (OTT) compared to ours (40 days vs. 10 days). Zaorsky et al., in their meta-analysis, found that brachytherapy regimens with an EQD2 above 50 Gy were associated with improved local control but a slightly higher risk of late toxicity. Guinot et al. recommended an EQD2 of at least 50 Gy for optimal tumor control but also highlighted that regimens delivering EQD2 values between 45 and 50 Gy were associated with excellent cosmetic outcomes [21]. We opted for a slightly lower dose as most patients were individuals who, for esthetic reasons, had declined a surgical approach. Consequently, our primary objective was to minimize the risk of potential side effects while ensuring effective treatment.

The lower per-fraction dose of 3 Gy BID ensures a more gradual dose accumulation, reducing the risk of fibrosis, telangiectasia, and mucosal ulceration, which are common concerns in mucosal irradiation, which is why some authors suggest a limit of 3 Gy per fraction as the maximum acceptable cut-off to achieve better safety and cosmetic results [14]. Furthermore, the BID fractionation could help counteract tumor repopulation, particularly in histological subtypes with aggressive biological behavior.

Moreover, except for two cases, our series primarily utilized interstitial or endocavitary catheters, reflecting the presence of larger tumors that were not suitable for a purely superficial approach and whose treatment could have been associated with an increased risk of toxicity.

Based on the above considerations, we believe that our regimen, while not among the most used, could be an accepted and justifiable approach. As further surgery was intentionally excluded in agreement with patients to prioritize cosmetic outcomes, particularly for lesions in the midface region where esthetic considerations are paramount, a less aggressive but still effective radiation regimen was deemed the most appropriate.

At every stage of treatment, including—especially—target indication and delineation, close collaboration among multiple specialists is highly desirable, if not essential, to ensure adequate coverage of high-risk areas. This is particularly crucial in these malignancies, which are often superficial and may not be well visualized on imaging, making it imperative to delineate the target from multiple perspectives. In this context, a multidisciplinary approach involving a radiation oncologist, dermatologist, and otolaryngologist is fundamental. Further research with larger cohorts and a longer follow-up is warranted to establish standardized treatment guidelines.

Multidisciplinary cooperation is mandatory even in the implantation phase, especially when interstitial implants are indicated. Although most series have been performed using a superficial approach [23,24,25]—which naturally limits the range of lesion size and depth that can be effectively treated—the literature includes multiple retrospective reports evaluating interstitial brachytherapy for facial cutaneous tumors [26,27,28]. A noteworthy series by Rio et al. [29] retrospectively analyzed 97 skin carcinomas (88 BCC cases, 9 SCC cases) of the nose, periorbital areas, and ears, treated with interstitial catheters—similar to the technique adopted in most of our cases. Despite using slightly higher doses then ours (mean 55 Gy vs. our study’s 44–45 Gy), they reported comparable outcomes, with a local control rate of 92.5% and good cosmetic results.

In the present series, only two cases required interstitial implantation, while we resorted to endocavitary implants in most cases. To our knowledge, this is the first reported use of endocavitary implants integrated into nasal packing material for treating nasal BCC. When used alongside surface implants, endocavitary techniques offer a key advantage in high-risk cases: they enable comprehensive coverage of the nasal wall’s full thickness—a task that would traditionally demand interstitial implantation. Additionally, these implants can typically be placed in outpatient settings without anesthesia or an operating room, providing a distinct practical advantage over interstitial methods. The personalized combination of interstitial, contact, and endocavitary delivery is, to our knowledge, an innovative approach which can increase the possibilities and indications of brachytherapy for the treatment of bulky nose lesions for which the cosmetic impact of surgery is expected to be the most relevant.

Despite the promising outcomes observed in this study, certain limitations must be acknowledged.

First, the retrospective nature of this study introduces potential biases, particularly in patient selection and outcome assessment. A prospective, randomized comparison with alternative radiotherapy modalities or surgical approaches would strengthen the evidence supporting the use of our fractionation regimen.

Second, the relatively short follow-up period limits our ability to fully assess long-term disease control and late toxicities. While our results indicate high local control rates and excellent cosmetic outcomes in the short to mid term, a longer follow-up would provide a more comprehensive evaluation of late recurrences, fibrosis, and chronic radiation effects.

Lastly, this study focuses on a single-institution experience, limiting the generalizability of our findings. Multicenter studies with larger sample sizes would be necessary to confirm the reproducibility of our results and establish optimal treatment guidelines for HDR-IRT in facial BCC.

Despite the increasing clinical application of HDR-IRT for skin cancers, including BCC, current evidence is still largely derived from retrospective studies and single-institution experiences. This results in a lack of standardization in fractionation schedules, dosimetric parameters, and toxicity reporting. To achieve more robust and generalizable evidence, it is essential to promote data sharing and develop large-scale, multi-institutional databases that can provide high-quality evidence for treatment optimization.

Encouraging participation in multi-institutional registries or in prospective studies could significantly enhance the quality of evidence available and support the development of consensus guidelines that improve patient outcomes while maintaining optimal cosmetic and functional results.

## 5. Conclusions

The current and available data do not support brachytherapy as an equivalent oncologic alternative to surgery given that current evidence on HDR-IRT remains limited and is based solely on retrospective studies; rather, these data should be considered as a proof of concept. However, HDR-IRT represents a safe, effective, and cosmetically favorable option for treating facial BCC, particularly in elderly patients or those for whom surgery is contraindicated. Our institutional experience confirms that the local control rate of 87.5% is consistent with the findings in the literature and that cosmetic outcomes are highly favorable with limited toxicity. Endocavitary implantation, usually not employed in skin lesions, is a rational approach used to cover the full thickness of the nasal wall in high-risk cases.

Treatment was well tolerated, with excellent local control and patient compliance.

Future research should focus on long-term follow-ups, dosimetric analysis, and direct comparisons with external beam radiotherapy and surgery, possibly in prospective and randomized trials, in order to establish HDR-IRT as a standard treatment modality. However, our findings contribute to the growing body of evidence suggesting that IRT could become a standard of care for selected BCC cases, particularly when surgery is not a viable option.

## Figures and Tables

**Figure 1 jcm-14-03305-f001:**
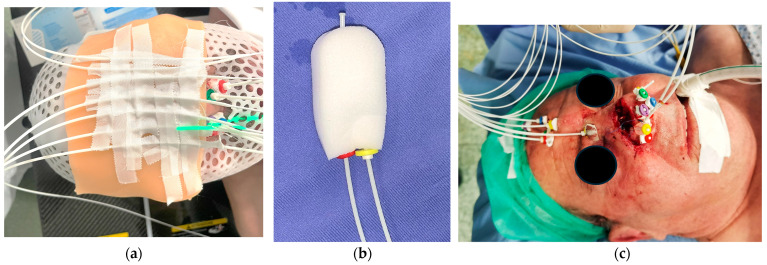
Types of implants. (**a**) Surface implant: a thermoplastic mask with plastic tubes placed on top of a bolus, following parallel lines that can adequately cover the surface of the target; (**b**) endocavitary implant: intranasal packing used for endocavitary implants with two plastic tubes positioned inside; (**c**) an interstitial implant with the final geometry of the implants being dictated by the anatomic subchondral planes.

**Figure 2 jcm-14-03305-f002:**
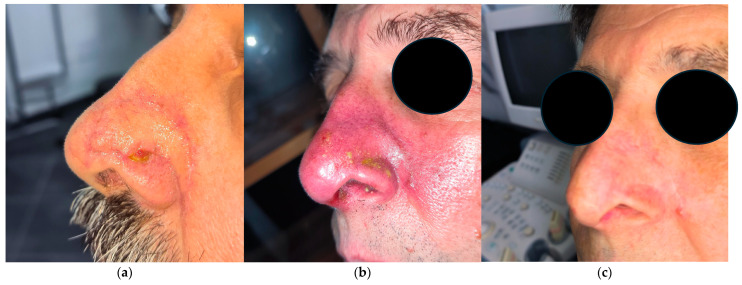
Male patient, 62 years old, with recurrent basal cell carcinoma previously treated with multiple cycles of cryotherapy and R1 excisions, presenting, before treatment, with disease recurrence on left nasal ala. (**a**) Before treatment; (**b**) two weeks after end of treatment; (**c**) two months after end of treatment.

**Table 1 jcm-14-03305-t001:** Patient characteristics.

#	Age at Time of Treatment	Sex	Site	Main Involved Side	Hystological Subtypes	Type of Disease
1	68	M	Nose	Right	Nodular	Primary
2	57	F	Upper Lip	Right	Morpheaform	Primary
3	62	M	Nose	Left	Nodular	Recurrent
4	72	M	Nose	Left	Infiltrative	Recurrent
5	90	M	Nose	Left	Nodular	Recurrent
6	84	M	Nose	Left	Infiltrative	Recurrent
7	59	M	Nose	Left	Nodular	Primary
8	77	M	Nose	Right	Basosquamous	Recurrent

Abbreviations: M: male; F: female.

**Table 2 jcm-14-03305-t002:** Treatment characteristics.

#	CTV_cc	CTV Dmean (Gy)	Dose Total (Gy)/#Fractions	Total Number of Plastic Tubes	Endocavitary Tubes	Surface Tubes	Interstitial
1	1.08	62.3	44/14	5	0	5	0
2	13.3	93.5	45/9	5	0	0	5
3	6	53.08	44/14	7	2	5	0
4	4.1	59.05	44/14	9	2	7	0
5	11.54	55.05	44/14	8	2	6	0
6	3.71	62.7	44/14	4	0	4	0
7	18.92	76.06	44/14	11	4	7	0
8	23.28	78.7	44/14	8	2	0	6

Abbreviations: CTV: Clinical Target Volume.

## Data Availability

The data presented in this study are available on request from the corresponding author.

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
