# Peer review of "Preserving Esthetics: Interventional Radiotherapy (Brachytherapy) as a Potential Alternative to Surgery for Basal Cell Carcinoma of the Midface"

_jcm, 2025, doi:10.3390/jcm14103305_

Round 1

Reviewer 1 Report

Comments and Suggestions for Authors

Re et al. present retrospective, single-center data on the use of High-Dose Rate Interventional Radiotherapy (HDR-IRT) for treating basal cell carcinoma (BCC). The authors highlight that BCC is one of the most common forms of skin cancer, frequently occurring in sun-exposed areas such as the midface—an anatomically and cosmetically sensitive region. Having previously published their expertise in HDR-IRT for other cancers, such as nasal vestibule cancer, the authors now explore its application for BCC, aiming to achieve effective local control with favorable aesthetic outcomes.

A total of eight patients were analyzed.

The article is well-written and engaging. However, the following points should be addressed:

  1. Patient Number in the Abstract – The total number of patients should be explicitly stated in the abstract.

  2. Cosmetic Outcome Details – The paper frequently emphasizes the good cosmetic outcomes of HDR-IRT. Since this is a key argument supporting the treatment, further details would enhance the discussion. For example, was a patient-reported questionnaire used, or were outcomes assessed solely by physicians? How many patients experienced any form of cosmetic side effects, such as scarring?

  3. Description of Toxicity – The study states that side effects were “minimal,” yet 2 out of 8 patients experienced Grade III toxicities. A more accurate phrasing, such as “limited toxicities,” would be preferable.

A key limitation of this study is the small sample size and short follow-up period. It is important to emphasize that the available data do not support brachytherapy as an equivalent oncologic alternative to surgery, given that current evidence on HDR-IRT remains limited and is based solely on retrospective studies. However, given the innovative approach—particularly the use of endocavitary implants.

Author Response

Report 1

Comments and Suggestions for Authors

Re et al. present retrospective, single-center data on the use of High-Dose Rate Interventional Radiotherapy (HDR-IRT) for treating basal cell carcinoma (BCC). The authors highlight that BCC is one of the most common forms of skin cancer, frequently occurring in sun-exposed areas such as the midface—an anatomically and cosmetically sensitive region. Having previously published their expertise in HDR-IRT for other cancers, such as nasal vestibule cancer, the authors now explore its application for BCC, aiming to achieve effective local control with favorable aesthetic outcomes.

A total of eight patients were analyzed.

The article is well-written and engaging. However, the following points should be addressed:

  1. Patient Number in the Abstract – The total number of patients should be explicitly stated in the abstract. 

Thank you, we state the number of patients in the abstract

  1. Cosmetic Outcome Details – The paper frequently emphasizes the good cosmetic outcomes of HDR-IRT. Since this is a key argument supporting the treatment, further details would enhance the discussion. For example, was a patient-reported questionnaire used, or were outcomes assessed solely by physicians? How many patients experienced any form of cosmetic side effects, such as scarring?

We appreciate this valuable suggestion and agree that a more detailed description enhances the understanding of our findings. As requested, we have expanded this section in the Results, adding a paragraph that describes in greater detail the nature and timing of the observed cosmetic effects, as well as the methods used for clinical assessment. We hope this addition provides a clearer picture of the favorable cosmetic profile of the treatment.

  1. Description of Toxicity – The study states that side effects were “minimal,” yet 2 out of 8 patients experienced Grade III toxicities. A more accurate phrasing, such as “limited toxicities,” would be preferable.

Thank you for highlighting this potential source of confusion. We have revised the sentences and the definition of toxicity accordingly to improve clarity and avoid misinterpretation, following your recommendation.

A key limitation of this study is the small sample size and short follow-up period. It is important to emphasize that the available data do not support brachytherapy as an equivalent oncologic alternative to surgery, given that current evidence on HDR-IRT remains limited and is based solely on retrospective studies. However, given the innovative approach—particularly the use of endocavitary implants.

Thank you very much for the comment. We agree and in the new version of the manuscript, we explicitly listed these and other limitations, and in the conclusions we outline that brachytherapy cannot yet considered as an equivalent oncologic alternative to surgery.

Reviewer 2 Report

Comments and Suggestions for Authors

Abstract:

Well-structured abstract, containing all the necessary elements for a summary presentation of the study conducted. I appreciate the clarity of the writing, which I believe can attract readers even less familiar with the field.

The keywords are well chosen, necessary and useful to this manuscript.

On a scale of 1 to 10, I’ll give 10 points for this abstract.

Introduction:

The introduction is interesting, presents in sufficient detail the treatment issues that this type of cancer and this location raise. As in the abstract, the clear language and sufficiently detailed explanations I believe can broaden the audience of this manuscript. I have no objections to this chapter.

On a scale of 1 to 10, I agree 10 points for this introduction.

Methodology:

The methodology is presented correctly and convincingly, accompanied by suggestive images. The impression it leaves is that, under similar conditions, the study can be reproduced, and I believe that this is the essential object that the description of the methodology must achieve.

On a scale of 1 to 10, I must agree 10 points for methodology.

Results:

The results chapter is a bit unbalanced. There are only about 20 lines of text, which I kept rereading...it seems a bit too little to me. I think this chapter deserves a bit more. The 2 tables and 3 pictures are welcome.

On a scale of 1 to 10, I agree 8 points for results.

Discussion:

The discussions have both pluses and minuses. The chapter is consistent, but relies on an impermissibly small number of bibliographical references. I cannot believe that the topic under discussion has been so little present in the recent literature.

The limitations of the study are presented in detail and I commend the authors for their honesty.

However, right from the introduction, the authors also promised a review of the literature (I quote: “This review combines our experience with an analysis of the literature…”). I have now to ask two questions.

Where is the description (even briefly) of the methodology used to conduct this literature review?

Then, how was it possible for a literature review to be limited to only 16 bibliographic references?

In this situation, on a scale of 1 to 10, I agree only 5 points for discussion.

Conclusion:

The conclusions are well written in relation to the manuscript, I have no comments on them.

On a scale of 1 to 10, I will give 9 points for conclusions.

Bibliography/References:

16 bibliographic references correctly drafted and correctly cited in the text, in my opinion represents far too little for any type of manuscript, let alone one that also wants to be a literature review. The impression it leaves on the reader is that the authors did not do enough research, and, frankly, I doubt that is the case.

On a scale of 1 to 10, I agree 5 points for the bibliography.

Figures/Tables:

In terms of graphics, 2 tables and 2 figures illustrate this manuscript quite well. If I were the authors, I would consider supplementing the graphics; it would make the manuscript even more attractive.

On a scale of 1 to 10, I agree 8 points for this chapter.

Review Decision:

Reconsider after major revisions.

Author Response

Report 2
Comments and Suggestions for Authors
Abstract: Well-structured abstract, containing all the necessary elements for a summary presentation of the study conducted. I appreciate the clarity of the writing, which I believe can attract readers even less familiar with the field. 
The keywords are well chosen, necessary and useful to this manuscript.
On a scale of 1 to 10, I’ll give 10 points for this abstract.
Thank you very much.
Introduction: The introduction is interesting, presents in sufficient detail the treatment issues that this type of cancer and this location raise. As in the abstract, the clear language and sufficiently detailed explanations I believe can broaden the audience of this manuscript. I have no objections to this chapter. On a scale of 1 to 10, I agree 10 points for this introduction.

Thank you very much.
Methodology: The methodology is presented correctly and convincingly, accompanied by suggestive images. The impression it leaves is that, under similar conditions, the study can be reproduced, and I believe that this is the essential object that the description of the methodology must achieve. On a scale of 1 to 10, I must agree 10 points for methodology. 
Thank you very much.
Results: The results chapter is a bit unbalanced. There are only about 20 lines of text, which I kept rereading...it seems a bit too little to me. I think this chapter deserves a bit more. The 2 tables and 3 pictures are welcome. On a scale of 1 to 10, I agree 8 points for results.
Thank you very much, you are right, in the new version we expanded the results section as suggested.
Discussion: The discussions have both pluses and minuses. The chapter is consistent, but relies on an impermissibly small number of bibliographical references. I cannot believe that the topic under discussion has been so little present in the recent literature. The limitations of the study are presented in detail and I commend the authors for their honesty. 
However, right from the introduction, the authors also promised a review of the literature (I quote: “This review combines our experience with an analysis of the literature…”). I have now to ask two questions.
Where is the description (even briefly) of the methodology used to conduct this literature review?
Then, how was it possible for a literature review to be limited to only 16 bibliographic references?
In this situation, on a scale of 1 to 10, I agree only 5 points for discussion.
Thank you very much, you are right, on one hand we better described, at the end of the introduction, the aim of the study which is not a systematic review of the literature, on the other hand we integrated our revision of existing evidence on the matter and expanded the discussion accordingly thus adding 14 works to the bibliography.
Conclusion: The conclusions are well written in relation to the manuscript, I have no comments on them.  On a scale of 1 to 10, I will give 9 points for conclusions.
Bibliography/References: 16 bibliographic references correctly drafted and correctly cited in the text, in my opinion represents far too little for any type of manuscript, let alone one that also wants to be a literature review. The impression it leaves on the reader is that the authors did not do enough research, and, frankly, I doubt that is the case. On a scale of 1 to 10, I agree 5 points for the bibliography. 
Thank you very much, you are right, on one hand we better described, at the end of the introduction, the aim of the study which is not a systematic review of the literature, on the other hand we integrated our revision of existing evidence on the matter and expanded the discussion accordingly thus adding 14 works to the bibliography.
Figures/Tables: In terms of graphics, 2 tables and 2 figures illustrate this manuscript quite well. If I were the authors, I would consider supplementing the graphics; it would make the manuscript even more attractive. On a scale of 1 to 10, I agree 8 points for this chapter.
Thank you very much.

Reviewer 3 Report

Comments and Suggestions for Authors

Thank you for submitting this interesting pilot study that aimed to investigate the clinical outcomes of patients with basal cell carcinoma of mid-face (n=8) who underwent HDR-IRT at a single  institution with short follow up.

The authors concluded that HDR-IRT outcomes were favourable and it was a well-tolerated, effective, and cosmetically favorable treatment for midface BCC.

The manuscript was well written and text was clear and easy to read

Unfortunately although the study population was interesting, the study was weak and limited by serious factors:

1- The small size  of the study cohort from a single institution with short follow up

2- The study cannot be generalised.

3- lacking novelty and innovation

4- The retrospective nature of the study and lacking for control group and controlling for confounders.

5- The heterogeneity of the study population and the high potential of selection biases

6- Overfitting of the results

7- Lacking of the external validation

8- No control for type 1 and 2 statistic errors.

9- The authors did not addressed in details the future plans and directions to validate their preliminary results and how to validate the clinical utility of the study.

Author Response

Report 3
Comments and Suggestions for Authors: Thank you for submitting this interesting pilot study that aimed to investigate the clinical outcomes of patients with basal cell carcinoma of mid-face (n=8) who underwent HDR-IRT at a single institution with short follow up.
The authors concluded that HDR-IRT outcomes were favourable and it was a well-tolerated, effective, and cosmetically favorable treatment for midface BCC.
The manuscript was well written and text was clear and easy to read.Unfortunately although the study population was interesting, the study was weak and limited by serious factors:
1- The small size  of the study cohort from a single institution with short follow up.
Thank you very much we agree and acknowledged this limitation in the discussion
2- The study cannot be generalised.
We agree and acknowledged this limitation in the discussion adding it as a limitation.
3- lacking novelty and innovation.
Thank you very much, we do believe that this is in part true and in the text we now try to better outline the novelty and innovation potential. In particular we stressed that “the personalized combination of interstitial, contact and endocavitary delivery is to our knowledge an innovative approach which can increase the possibilities and indications of brachytherapy for the treatment also of bulky nose lesions for which the cosmetic impact of surgery is expected to be most relevant.”
4- The retrospective nature of the study and lacking for control group and controlling for confounders.
Thank you very much we agree and acknowleged this while listing the limitations.
5- The heterogeneity of the study population and the high potential of selection biases.
Thank you very much we agree and acknowledged this among the limitations
6- Overfitting of the results.
We agree, we better state in the new version of the manuscript that the aim is to describe our particular experience, opening new perspectives for validation of this approach, and that the present work shoul be intended more as a proof of concept than as a definite validation of a treatment modality.
7- Lacking of the external validation.
Thank you very much we agree and acknowledged this among the limitations
8- No control for type 1 and 2 statistic errors.
Thank you very much we agree and acknowledged this among the limitations, deriving also from the small sample as it is more a proof of concept than a fully conclusive report.
9- The authors did not addressed in details the future plans and directions to validate their preliminary results and how to validate the clinical utility of the study.
Thank you very much we agree and in the conclusion we added the following sentence: “Future research should focus on long-term follow-up, dosimetric analysis, and direct comparisons with External Beam Radiotherapy and surgery, possibly in prospective and randomized trials, in order to further establish HDR-IRT as a standard treatment modality”.

Round 2

Reviewer 3 Report

Comments and Suggestions for Authors

Thank you for submitting this revised manuscript and as the authors changed the aim of the study to a prof of concept that would pave the way for further investigation in larger cohort and addressed the limitations of the study in more details in the discussion section, I found the manuscript is suitable for publication.